# Focus on Translation Initiation of the HIV-1 mRNAs

**DOI:** 10.3390/ijms20010101

**Published:** 2018-12-28

**Authors:** Sylvain de Breyne, Théophile Ohlmann

**Affiliations:** 1CIRI, Centre International de Recherche en Infectiologie, 46 Allée d’Italie, 69364 Lyon, France; theophile.ohlmann@ens-lyon.fr; 2Univ Lyon, 46 Allée d’Italie, 69364 Lyon, France; 3Inserm U1111, 46 Allée d’Italie, 69364 Lyon, France; 4Université Claude Bernard Lyon 1, 46 Allée d’Italie, 69364 Lyon, France; 5CNRS UMR5308, 46 Allée d’Italie, 69364 Lyon, France; 6ENS de Lyon, 46 Allée d’Italie, 69364 Lyon, France

**Keywords:** HIV-1, unspliced mRNA, translation initiation, RNA helicases, IRES

## Abstract

To replicate and disseminate, viruses need to manipulate and modify the cellular machinery for their own benefit. We are interested in translation, which is one of the key steps of gene expression and viruses that have developed several strategies to hijack the ribosomal complex. The type 1 human immunodeficiency virus is a good paradigm to understand the great diversity of translational control. Indeed, scanning, leaky scanning, internal ribosome entry sites, and adenosine methylation are used by ribosomes to translate spliced and unspliced HIV-1 mRNAs, and some require specific cellular factors, such as the DDX3 helicase, that mediate mRNA export and translation. In addition, some viral and cellular proteins, including the HIV-1 Tat protein, also regulate protein synthesis through targeting the protein kinase PKR, which once activated, is able to phosphorylate the eukaryotic translation initiation factor eIF2α, which results in the inhibition of cellular mRNAs translation. Finally, the infection alters the integrity of several cellular proteins, including initiation factors, that directly or indirectly regulates translation events. In this review, we will provide a global overview of the current situation of how the HIV-1 mRNAs interact with the host cellular environment to produce viral proteins.

## 1. Introduction

Flexibility and adaptability are both essentials when facing difficult situations; such an assumption retains all its meaning when it comes to the viral kingdom. Indeed, viruses, unlike other pathogens, such as bacteria, require a host to replicate and spread. One of the key steps of the viral cycle, which is also completely dependent on the host, is protein synthesis and always occurs with cellular ribosomes. In a very interesting way, the host’s innate immune response, the first line of defense against viral infection, targets the actors of translation to stop protein synthesis. In addition, most of viral messenger RNAs (mRNAs) contain many different RNA motifs that are essential for replication but are not compatible with a conventional mechanism of translation. Therefore, viruses have to modify and adapt to produce proteins by implementing different strategies to circumvent the host cell’s defenses and the RNA barriers.

The type 1 human immunodeficiency virus (HIV-1) is a member of the *Lentivirus* genus from the *Retroviridae* family and the etiologic agent of the acquired immunodeficiency syndrome. It is a good representative of this viral adaptability with more than 50 viral RNA transcripts produced by alternative splicing [1,2,3] that allow for the expression of 15 proteins [4].

Through this review, we will first and briefly describe the mechanisms by which some viral elements intervene to suppress activation of the first line of defense against viral infections at the translational level. Next, we will see that alternative splicing of the primary transcript generates a multitude of mRNAs that contribute to regulate the translational efficiency. Lastly, we will focus on the unspliced genomic RNA. Indeed, translation of this transcript can use both a classical translation initiation mechanism and internal recruitment of ribosomes in order to adapt to the cellular modifications imposed by infection.

## 2. The Diversity of Translation Initiation Mechanisms

In higher eukaryotes, translation initiation requires association of the mRNA to the ribosomal subunits. This event is orchestrated in a timely manner by a set of proteins that allows progression through the different stages. Initiation begins with the attachment of the small 40S ribosomal subunit with associated factors to the mRNA and progression of the ribosome in a 5′ to 3′ direction to find and recognize the initiation codon. The elongation stage allows peptide bond formation in the ribosomal peptidyl transferase center between incoming amino acids and the nascent peptide until ribosomes reach a stop codon. This codon arrests protein synthesis and induces the release of the neo-synthesized peptide. Recycling of the ribosomal subunits constitutes the last and final step of protein synthesis. The initiation step is complex and highly regulated by at least 12 eukaryotic initiation factors (eIFs), which catalyze several processes such as the binding of the initiator methyonyl-transfert RNA (Met-tRNAi) to the small ribosomal subunit, ribosomal recruitment at the 5′end, ribosomal scanning, and initiation codon recognition. Translation efficiency is modulated by both RNA structures, RNA modifications, and by the interplay with eIFs and a large number of cellular proteins. Each mRNA is unique in its folding and modifications, which requires an adaptation of ribosome recruitment to these characteristics. Thus, it does not come as a surprise that translation initiation can be mediated by several different mechanisms. [5]

### 2.1. Cap-Dependent Translation Initiation

#### 2.1.1. The Canonical Scanning Model

All eukaryotic mRNAs contain an N7-methylated guanosine linked to the first nucleotide of the RNA via a reverse 5′ to 5′ triphosphate linkage called the cap structure, which is required for mRNA processing and nuclear export, but also for stability and translation efficiency [6].

Most mRNAs initiate translation via a cap dependent and scanning mechanism (Figure 1), which starts by the binding of the small ribosomal subunit on the 5′ cap. This requires a set of three factors composed of eIF4E, eIF4A, and eIF4G that form the eIF4F complex. The mRNA cap structure is specifically recognized by the eIF4E cap binding protein. eIF4E then binds eIF4G, which is often named the scaffold protein as it maintains the architecture of the complex by contacting multiple interactions with other partners such as eIF4A, eIF3, and the poly(A)-binding protein (PABP). The ATP dependent (Asp-Glu-Ala-Asp) DEAD-box RNA helicase eIF4A is directly implicated in the RNA structures’ unwinding that are often found close to the 5′ end. Thus, the eIF4F complex can activate the mRNA to allow proper recruitment and landing of the small ribosomal subunit at the cap structure [7,8]. Activity of the eIF4F complex is enhanced by the presence of eIF4B and eIF4H that assist RNA unwinding during ribosomal scanning [9]. In addition, eIF4G also interacts with PABP which binds to the poly(A) tail located at the opposite end of the mRNA to allow the pseudo-circularization of the transcript in a closed-loop conformation [10].

Initiation generally occurs at an AUG codon, which is recognized by the Met-tRNAi with perfect codon/anticodon annealing. The Met-tRNAi has a high affinity for eIF2●GTP and forms the ternary complex (TC). TC binding to the 40S subunit is strongly enhanced by eIF1, eIF1A, and eIF3 to form the 43S pre-initiation complex (PIC). eIF3, which is the largest of all initiation factors, is composed of 13 subunits and interacts with eIF4G to bridge the 43S PIC to the mRNA cap-binding complex [11,12,13]. Once attached on the mRNA, the PIC needs to localize the AUG codon along the 5′ untranslated region (UTR). This generally occurs via the linear scanning of the ribosome on the 5′ UTR and requires cellular RNA helicases such as eIF4A and DHX29 to unwind the potential extensive RNA secondary structures [14,15]. Usually, the first AUG codon read by the 43S PIC is used to initiate translation. However, the nucleotide context influences the initiation rate and the sequence GCCA/GCC**AUG**G (referenced as the “Kozak sequence”) has been reported to favor start codon recognition [16,17]. Once the 43S PIC is localized onto the AUG, this triggers hydrolysis of the GTP bound to eIF2 which, in turn, causes conformation modification of the PIC with dissociation of initiation factors. Then, eIF5B mediates the association of the 60S ribosomal large subunit to the 40S into an 80S ribosome and marks the end of the initiation phase. The reader can find more details on translation initiation in these excellent following reviews: References [7,8,18,19].

#### 2.1.2. Initiation by Leaky Scanning

Most of the time, the first AUG codon encountered during ribosomal scanning of the 5′ UTR is used to initiate translation. However, the efficiency of codon selection depends on several parameters. Hence, the nucleotides sequence surrounding the initiation codon, called the “Kozak context” [16,17], plays an important role for the recognition of the initiation site, especially positions −3 and +4 that flank the AUG [20]. Nucleotide variation at these positions contributes to create a weak Kozak context that decreases the start codon recognition efficiency. This is even more pronounced when eIF2α is phosphorylated and less available to initiate (see Section 2.3). Thus, the first AUG codon can be ignored by the scanning 43S PIC, which continues to scan the 5′ UTR further downstream to locate and find the next AUG codon [8]. This mechanism, called leaky scanning [21,22], allows the production of two isoforms of the same protein if located in the same reading frame [23,24], or two completely different proteins from two distinct open reading frames (ORFs) [24]. 

### 2.2. Cap-Independent Translation Initiation

#### 2.2.1. Internal Ribosome Entry Segments (IRESes)

IRESes have been first identified in the genomic RNA of two members of the picornaviral family in the late eighties (poliovirus and encephalomyocarditis virus (EMCV)) [25,26]. Picornaviral infections are associated with a strong shut-off of cellular translation without affecting viral mRNA expression. This repression is the consequence of virally encoded proteases that cleave the initiation factors eIF4G, PABP, or eIF5B [27,28,29,30]. Interestingly, picornaviral mRNAs are uncapped and they harbor an unusually long and structured 5′ UTR with several silent AUG codons. Despite these structural features, translation of these picornaviral mRNAs remains very efficient during the loss of function of eIF4G, PABP, and eIF5B. This is explained by the fact that viral translation occurs by an alternative mechanism independent of the cap structure that requires the appropriate folding of the RNA structures to interact directly with components of the 43S PIC (Figure 1). These RNA elements constitute an IRES. Unlike cap-dependent initiation, IRESs mediate translation initiation with a limited subset of eIFs and have been now identified in different viral families such as *Picornaviradae*, *Flaviviradae*, *Dicistroviridae*, and *Retroviridae* [31], and in many cellular mRNAs [32]. The mechanism of viral IRESs has been extensively investigated leading to a classification according to their requirement in eIFs (Table 1 and Figure 1) [31,33]. IRESs mediated translation often provides viral mRNAs with a selective advantage as translation is sustained in circumstances where cap-dependent translation is altered by cellular events (see Section 2.3) or during viral infection (see Section 5.2.1).

#### 2.2.2. N6-Methyladenosine-Induced Ribosome Engagement Site 

N6-methyladenosine (m^6^A) on mRNA is a post-transcriptional modification mediated by a regulated interplay between m^6^A methyltransferases (named writer), demethylase (named eraser) and m^6^A-specific binding protein (called reader) [45]. The methyltransferase like 3 (METTL3) associated with cofactors including METTL4 and Wilms’ tumor 1-associating protein (WTAP) [46,47,48] forms a complex that interacts with the mRNA, mostly in the nucleus, to promote m^6^A. This modification is reversible by at least two eraser proteins, which are FTO and ALKBH5 (for fat mass and obesity-associated and AlkB homolog 5 respectively) [49,50]. There is only a minor fraction, 0.1% to 0.5%, of adenosine residues that are modified per mRNA [51,52] with a preference for the consensus sequence RRACU (R = A or G) [53]. This occurs mostly in the coding sequence and in the 3′ UTR [54], and controls mRNA stability and translation [45]. The reader human YTH domain family (YTHDF) proteins can have opposite functions when they interact with m^6^A. YTHDF1 is associated with a stimulation of the translation efficiency of its mRNA target while YTHDF2 promotes mRNA degradation [55]. The mechanism that allows this increase of translation is not completely understood, but it is known that YTHDF1 interacts with eIF3 [55]. In a same way, tethering in the 3′ UTR of the METTL3 writer protein, which catalyzes m^6^A modification, enhances translation of the target mRNA [56,57]. This increase of translation is dependent on the interaction between METTL3 and the h subunit of eIF3 [57]. These data suggest that the m^6^A residues located in 3′ UTR promotes mRNA circularization, via an interaction YTHDF1/eIF3 or METTL3/eIF3, that enhances translation. 

A subset of mRNAs contains m^6^A residues in their 5′ UTR [54] that allow a m^6^A-induced ribosome engagement site (MIRES, Figure 1) [58]. Silencing of the expression of the METTL3 writer results in a strong and specific reduction of both m^6^A residues located in the 5′ UTR and the decrease of translation efficiency [58]. Ribosomal complex formation on m^6^A mRNAs has been analyzed by toeprinting assay and does not require any components of the eIF4F complex indicating that m^6^A can support cap-independent translation. Consistent with this observation, translation of m^6^A mRNAs is robust in HeLa cell extracts with low activity of eIF4E [58]. In order to identify cellular partners that mediate ribosomal recruitment, Meyer et al have used a UV-crosslinking assay and showed that mRNAs bearing an m^6^A are directly and preferentially associated with eIF3 independently of the YTHDF1 protein [58]. Coots et al. have also shown that translation of mRNAs with m^6^A residues in the 5′ UTR are independent from eIF4F but are sensitive to METTL3 repression [59]. Both studies support the hypothesis that translation mediated via m^6^A occurs independently from the cap and the cap-binding proteins (Figure 1). 

### 2.3. Regulation of Translation Initiation at the Level of eIFs

Proteins synthesis is highly regulated and is able to adapt quickly to physiological changes of the cell (differentiation, mitosis, apoptosis, and stress). A large part of these mechanisms of translation regulation occurs at the initiation step and focuses on two factors eIF2 and eIF4E. 

The activity of eIF2 is modulated by phosphorylation at serine 51 on the alpha subunit. As a consequence, the P-eIF2α binds with high affinity to the guanine nucleotide exchange factor eIF2B which reduces the recycling of inactive eIF2●GDP into active eIF2●GTP. The low amount of available and active eIF2●GTP in the cell inhibits the global level of translation [60,61]. Several kinases sensitive to various cellular stresses are involved in eIF2 phosphorylation including the interferon-inducible RNA-dependent protein kinase (PKR). This Ser/Thr kinase is a component of the first line of defense against viral infections, which is activated through auto-phosphorylation in the presence of viral double-stranded RNA (dsRNA) [62,63]. For instance, PKR is activated by HIV-1 transcripts, but the virus also develops several strategies to counteract the effects of this activation (see part 3 of this review).

The cap binding factor eIF4E is also regulated by phosphorylation or by sequestration by a family of small proteins that specifically bind to it and that are called 4E-binding proteins (4E-BP). Indeed, eIF4E binds to a protein motif that is common to both eIF4G and 4E-BP and the affinity of the interaction between eIF4E and 4E-BP is controlled by a series of phosphorylation events [64]. Hypophosporylation of 4E-BP strongly increases its affinity to eIF4E but reduces its availability to be incorporated in the eIF4F complex and results in a decrease of cap-dependent translation. However, when 4E-BP is hyperphosphorylated, eIF4E is released from its interaction with 4E-BP and can engage in translation with eIF4G. The phosphorylation status of 4E-BP is regulated by the mTOR signaling pathway [65,66] and eIF4E is phosphorylated on a serine at position 209 by the map kinase-interacting kinase (MNK). The role of this regulation remains controversial [67] with some studies indicating that P-eIF4E can promote initiation of some mRNAs whereas in vitro studies indicate that the binding of P-eIF4E to the cap structure is less efficient. Interestingly, eIF4G is able to interact directly with MNK suggesting a local regulation of eIF4E phosphorylation although the exact mechanism remains unclear [66]. Finally, most IRESs containing mRNAs do not require eIF4E to initiate translation and can sustain protein synthesis under conditions when the integrity of eIF4E is compromised. 

## 3. HIV-1 Escape from the Innate Immune Response at the Translation Level

The HIV-1 is a retrovirus that replicates via reverse transcription of its RNA genome into double-stranded DNA flanked by two long terminal repeats (LTRs). This proviral DNA is imported in the nucleus and integrated into the host genome. Then, all viral mRNAs are transcribed by the cellular type II RNA polymerase into capped and polyadenylated transcripts. As only one transcription start site exists on the HIV-1 genome, the ≈9700 bp primary viral transcript is alternatively spliced to generate a broad spectrum of viral mRNAs. All viral transcripts share the first 289 nucleotides (nt) located upstream of the 5′ major splice donor. The first 59 nt of HIV-1 form a very stable hairpin structure called the *trans*-activation response (TAR) element that plays a major role in stimulating the transcription of viral RNAs in the presence of *trans*-acting transcription (Tat) protein [68]. However, this dsRNA structure is also a strong signal for PKR activation [69,70,71,72]. However, during HIV-1 viral replication, PKR is activated only during the early phase of infection [69,70,71,72] and is quickly inhibited by viral and cellular factors. TAR is a binding site for several proteins including the viral Tat protein [73] and the cellular TAR RNA binding protein (TRBP) [74,75]. Interaction of these proteins with TAR sequesters the TAR hairpin and reduces the amount of RNA available for PKR activation [76,77]. TRBP also contributes to the inactivation of PKR through the formation of two heterodimers: TRBP/PACT (PACT protein is an activator of PKR [78]) and TRBP/PKR that reduces the activity of both PACT and PKR [79,80,81]. The dsRNA-specific adenosine deaminase 1 (ADAR1) is an interferon-stimulated gene that exhibits both antiviral and proviral functions [82]. ADAR1 and PKR interact with each other’s which lead to the inactivation of PKR. In the context of HIV-1-infected cells, the expression of ADAR1 is enhanced and the interaction between ADAR1 and PKR is favored, which allows viral RNA translation [71,72]. Surprisingly, PACT can also become an inhibitor of PKR during HIV-1 replication as its expression is stimulated and associated with multiprotein complexes composed of ADAR1, Tat, and TAR RNA that contributes to the inhibition of PKR phosphorylation [71,72,83,84]. For further details, we recommend the reader to consult these reviews: References [85,86,87].

## 4. HIV-1 Alternative Splicing Regulates the Level of Each Viral Transcript

Viral protein production depends on two main parameters: the intracytoplasmic concentration of viral mRNAs and their relative translational efficiency. The full-length unspliced genomic mRNA has two fates during viral replication: it serves as genomic RNA, which is encapsidated into new viral particles, and also plays the role of messenger RNA to produce the Gag and Gag-Pol polyproteins in the cytoplasm of the host cell. This ≈9700 bp primary viral transcript contains, at least, 5 donor and 8 acceptor sites used to generate more than 50 different transcripts (Figure 2) that can be extended to over 100 in some clinical isolates [2,88]. All these transcripts are then translated to produce the two major polyproteins (Gag and Gag-Pol), the two fragments of the envelope (Env), and six regulatory proteins (Tat, Nef, Rev, Vpu, Vpr, and Vif). Such discrepancy between the number of viral proteins and the viral transcripts indicates that viral expression is, in part, controlled at the level of alternative splicing. 

The primary transcript is fully spliced to generate 2 kb viral mRNA coding for Tat, Rev, and Nef [1]. They are the first ones produced during infection [89] and are exported to the cytoplasm via the nuclear RNA export factor 1 (NXF1)-mediated pathway similarly to host cellular mRNAs [90,91,92,93,94,95]. All other viral proteins are encoded from partially (4 kb) and unspliced transcripts (9.7 kb), which are expressed in a Rev-dependent manner. Indeed, Rev interacts both with the host Karyopherin chromosomal maintenance 1 protein (CRM1) [96,97] and the Rev-responsive element (RRE), which is present only on the 4 kb and 9 kb transcripts and located within the *env* gene (Figure 2) [98,99]. In addition, the human Staufen2, a double strand RNA binding protein, interacts in an RNA-independent manner with Rev to promote efficiently RNA export [100]. A fourth class of 1kb mRNAs has been described in a clinical isolate but its function has not been yet described [2].

The ratio of each viral transcript varies with the strength of the different splice sites which are regulated by an exonic splicing enhancer, silencer sequences, and by cellular proteins such hnRNP A/B/H and the SR proteins [3,88,101]. Hence, for the 2 kb mRNA family, the amount of transcripts coding for Tat is five times lower than mRNAs coding for Nef and Rev. Along the same line, mRNAs coding for Env/Vpu can represent up to 90% of the partially spliced mRNAs. These differences contribute to modulate the level of viral protein during replication.

An additional layer of complexity exists for each viral RNA species. Indeed, several transcripts can produce the same protein but they differ by the length and composition of their 5′ UTR. For instance, the Rev protein can be produced via 12 different isoforms of viral mRNAs expressed at a different ratio. A similar situation has been described for Nef, Tat, Vpr, and Env with 5, 8, 4, and 16 viral RNAs respectively [2,102]. This diversity is also due to the potential insertion of two non-coding exons (Figure 2). Surprisingly, translation initiation events on these mRNAs has never been thoroughly investigated but the multiplicity of viral transcripts certainly plays a key role as illustrated by the fact that mutations of the splice sites alter both the level of proteins produced and viral infectivity [1,103,104,105,106,107].

All these viral transcripts are temporally expressed in cells [89] because most of them are unspliced and required the viral Rev protein to be exported to the cytoplasm. The physiological status of the cell and the availability of the host translational machinery may differ between the early and late phases of viral replication (see Section 5.2.1) and this also modulates the translation of these different mRNAs. 

## 5. Translation of the HIV-1 Unspliced mRNA

The unspliced mRNA of HIV-1 drives translation of the two major polyproteins Gag and Gag-Pol. The 5′ UTR is 335 nt long (strain NL-4.3) and includes many structured RNA regions (Figure 3): (i) the TAR element is the binding site for the viral Tat protein which strongly stimulates transcription from the viral promoter [68], (ii) the polyadenylation signal (poly(A)) is present at both the 5′ and 3′ end of the viral transcripts but only the latter is involved in 3′ end processing [108], (iii) the primer binding site (PBS) contains the complementary sequence of the cellular tRNALys3 required for viral reverse transcription [109], (iv) the dimerization site (DIS) and the kissing loop hairpin are involved in the packaging of the two copies of genomic RNAs [110], (v) the 5′ splice donor (SD) site, and (vi) the packaging signals (Psi) which are critical for RNA splicing and encapsidation [111]. All these RNA motifs are absolutely essentials for viral replication, but they also constitute major hurdles for the scanning of the ribosomal initiation complex. The stable hairpin structure TAR element, located at the 5′ end of the messenger, forms the major obstacle to initiate via the conventional model [69,112,113,114,115]. However, HIV-1 has an incredible adaptability and has developed several strategies to initiate translation despite these structural constraints.

### 5.1. Cap- and Helicase-Dependent Scanning Model

The TAR structure is essential for *trans*-activation of viral RNA transcription and therefore for viral replication. However, the presence of this hairpin structure at the 5′ end of the unspliced HIV-1 transcript is an unequivocal barrier to ribosome recruitment at the cap. Indeed, insertion of TAR at the 5′ end of a reporter construct is sufficient to suppress RNA expression in in vitro translation systems like rabbit reticulocyte lysate (RRL) or Xenopus oocytes [69,112,113,114]. In addition, Parkin et al. have demonstrated that the accessibility of the cap structure was greatly reduced by the presence of TAR whereas mutations of nucleotides that destabilizes the base of the TAR stem loop can restore its accessibility [112]. Thus, the virus has developed strategies to overcome or to bypass the TAR structure and this occurs via the recruitment of cellular host proteins to promote viral translation (Figure 3). 

The La protein was originally identified as the major antigen for certain autoimmune diseases such as the systemic lupus erythematosus and the Sjögren’s syndrome [116]. Nuclear La protein is associated with nascent RNA polymerase III transcripts, while the cytoplasmic La protein participates in translational control of some viral RNAs [117,118]. Immunoprecipitation of the mRNP from HIV-1 infected lymphocytes revealed that La protein is associated with HIV-1 viral mRNA [119]. Specific interaction between La protein and the TAR RNA has been showed by mobility shift assay [119] and biacore analysis [120]. Addition of the recombinant La protein to the RRL is sufficient to relieve the translation inhibition imposed by the TAR structure [115]. In order to be active, the La protein requires its carboxy-terminal domain that promotes homodimerization and increases the expression of viral proteins [121]. The characterization of the function of La protein showed an ATP-dependent double-stranded RNA unwinding activity [122], which was suggested to promote TAR structure unwinding and the subsequent ribosomal recruitment at the cap structure. 

The cellular protein TRBP was discovered for its ability to bind the HIV-1 TAR RNA [75] and to inhibit activation of PKR (see Section 3) [85,87]. In addition to its role in the PKR mediated response, in vitro translation studies suggested that TRBP can also activate translation of a TAR-containing transcript via PKR-independent and uncharacterized mechanisms; this was confirmed for PKR-deficient cells [123]. These data indicate that TRBP could have a local function when bound to TAR RNA to promote cap-dependent translation of the unspliced mRNA. 

Staufen1 is a dsRNA-binding protein involved in multiple post-transcriptional gene-regulatory processes such as nuclear export, alternative splicing [124], mRNA transport in dentritic cells [125], mRNAs decay [126], and regulation of translation [127,128]. Using an RNA immunoprecipitation in tandem assay combined with high-throughput sequencing, Ricci et al. have shown that Staufen1 is associated with active translating ribosomes and interacts with mRNAs enriched in internal secondary structures [128]. Interestingly, Staufen1 also binds to the TAR RNA structure [127]. Addition of recombinant Staufen1 protein to the RRL or its overexpression into cells enhanced translation of TAR-containing transcripts, whereas silencing of Staufen1 specifically reduced their expression [127]. Dugré-Brisson et al. have also shown that Staufen1 particularly enhanced the association of TAR-containing RNAs with the polysomes [127]. Finally, Staufen1 was described to be associated with the RNA helicase A [129,130], although its role in RNA unwinding remains to be demonstrated. Interestingly, the Staufen family participates at different levels in the fate of the unspliced mRNA within the cell: from nuclear export to translation by the ribosome [100,127]. Due to these multiple roles, further investigation should be performed on this ds-RNA binding protein family in the future.

The RNA helicase A (RHA also named DHX9) is a highly conserved (Asp-Glu-Ala-Asp) DEAD-box protein involved in many different facets of cell metabolism such as RNA processing [131], innate anti-viral immunity [132], and in multiple steps of HIV-1 replication [133,134,135] including translation. RHA interacts with a sequence called post-transcriptional control element (PCE) in the HIV-1 5′ UTR, which overlaps the TAR element [133,136], and the binding of the helicase was shown to enhance viral transcription in the presence of Tat [133]. Silencing of RHA expression in cells reduces expression of an LTR-Gag plasmid [129,134] without altering the mRNA level [129], indicating that RHA modulates the translation initiation rate, probably via a mechanism that involves the unwinding of RNA secondary structures, although it has yet to be demonstrated.

The DEAD-box polypeptide 3 (DDX3) is a cellular ATP dependent RNA helicase whose expression is stimulated by the production of the Tat protein [137]. Meta-analysis of the human gene modifications that occur during HIV-1 infection has confirmed that expression of DDX3 was increased during HIV-1 replication [138]. Over-expression of DDX3 in cells enhances specifically the nuclear export and expression of RRE-containing RNAs in a Rev-dependent manner by interacting with CRM1 [137,139,140]. Curiously, silencing of DDX3 in cells does not suppress the export of unspliced mRNA indicating that the low amount of DDX3 still available could be sufficient to promote export; however, a strong impact on translation of the HIV-1 genomic RNA was reported [141,142]. We could further demonstrate a direct physical interaction between DDX3 and TAR, which allows the unwinding the viral stem-loop by the helicase. Interestingly, a series of gene reporters on which the TAR structure was displaced only 25 nt downstream to the cap structure were efficiently translated in DDX3 knocked-out cells. This indicates that DDX3 is needed to unwind RNA structures that are immediately adjacent to the cap structure [141]. Co-immunoprecipitation experiments from HeLa cells revealed an interaction between DDX3 and components of eIF4F complex, eIF3, and PABP [141,143]. In addition, DDX3 was retained on a m^7^GTP-sepharose column independently from eIF4E suggesting that the helicase could bind directly the 5′ cap structure of mRNAs [141,142]. All these data strongly suggest that DDX3 intervenes through several complexes including eIF4G, eIF4A and eIF3 to promote ribosomal recruitment at the cap structure in the environment of the TAR RNA structure (Figure 3). 

It is noteworthy that the proportion of active eIF4E is reduced during HIV-1 infection (see Section 5.2.1), leading to a strong inhibition of cellular mRNA translation, whereas HIV-1 transcripts are not affected [144,145]. The virus needs to adapt to these cellular modifications and has to bypass the use of eIF4E. Several mechanisms take place: IRESs (Section 5.2.2 and 5.2.5), adenosine methylation (Section 5.2.6), or subtle adaptation of the cap-dependent translation initiation. 

The nuclear cap binding complex (CBC) is composed of two cap binding proteins (CBP) CBP80 and CBP20 that interact with the 5′end of the mRNA into the nucleus and participate to the first round of mRNA translation in the cytoplasm [146]. Sucrose density gradients performed on cellular extracts transfected with HIV-1 coding plasmid indicated that CBP80 was associated with HIV-1 transcripts [145]. Recently, Toro-Ascuy et al have showed, using an in situ hybridization of probes directed to the unspliced mRNA coupled to a proximity ligation assay, that the HIV-1 genomic mRNA co-localized preferentially with CBP80 instead of eIF4E in the cytoplasm [147], which is consistent with previous work [145]. Over-expression in cells of CBP80, but not eIF4E, enhanced translation of the unspliced mRNA in a Rev-dependent manner and CBP80 and Rev were both found to be closely associated inside the nucleus [147]. In addition, the authors also showed that Rev is strongly associated to eIF4A1 and the recruitment of both eIF4A1 and CBP80 on the unspliced RNA is enhanced in the presence of both Rev and the RRE element [147], indicating that they are involved in translation initiation of these mRNAs (Figure 3). 

The unspliced HIV-1 mRNA also interacts with the human peroxisome proliferator-activated receptor-interacting protein with methyltransferase domain (PIMT) that is able to hypermethylate the guanosine of the cap structure to a trimethylguanosine (TMG), which is associated with an increase of unspliced mRNA export and translation [148]. This interaction is dependent on both the Rev protein and the RRE. Thus, it does not come as a surprise that the affinity of TMG cap structure for CBC is higher than eIF4E [149] and this supports the hypothesis that two complexes could be linked to an eIF4E-independent and scanning-dependent translation of the unspliced mRNA: the first one is composed of DDX3, eIF4G, and eIF4A [142], while the second is formed by CBP80, eIF4A1, and Rev [147]. These two complexes are both necessary to promote translation of HIV-1 transcripts, but their relative utilization remains to be demonstrated. However, we cannot exclude that these two complexes can internally bind to the 5′ UTR without any requirement for the cap. None of these authors have analyzed potential internal interaction of these proteins with the gRNA despite the fact that it is documented that Rev and DDX3 interact with the 5′ UTR [141,150,151]. 

To resume, many cellular helicases promote translation of the unspliced mRNA but their relative requirement during infection remains unclear. Do they co-operate and are they temporally regulated? Once recruited and attached to the mRNA, the PIC is able to scan the 5′ UTR through the RNA structures as demonstrated by artificial insertion of AUG codons in the 5′ UTR [152]. It was proposed that RNA architecture can regulate the flow of ribosomes engaged in translation on the unspliced mRNA. Consistent with this idea, the efficiency of the downstream ribosomal frameshifting [153] to produce Gag-Pol polyprotein is directly linked to the RNA structure in the 5′ UTR [154].

### 5.2. Alternative Mechanisms from the HIV-1 Cap-Dependent Initiation 

#### 5.2.1. HIV-1 Creates an Unfavorable Environment for Cap-Dependent Initiation

As we have seen, translation of the unspliced HIV-1 mRNA has to adapt to specific viral constraints such as (i) the export of unspliced transcripts via an alternative pathway, (ii) the presence of stable RNA structures at the 5 ‘end, and (iii) the requirement for non-canonical initiation factors. In addition, translation of the unspliced RNA takes place during the late stages of infection [89] when cellular conditions have been modified and are not favorable for cap-dependent translation. 

Structural modeling by selective 2′-hydroxyl acylation analyzed via primer extension (SHAPE) of the entire HIV-1 unspliced mRNA has confirmed that the 5′ UTR was highly structured [155]. Although ribosomal scanning takes place on the 5′ UTR [152] with the assistance of DDX3 [141,142], these RNA structures still constitute obstacles that may slow down the scanning of the ribosomes. Some studies have even showed that this low efficiency of translation imposed by the 5′ UTR was necessary for efficient ribosomal frameshifting to produce Gag-Pol [154]. 

The virally encoded HIV-1 protease, which is required for processing of Gag and Gag-Pol polyproteins during maturation, can also modulate cellular and viral translation. In the course of viral infection, this protease can cleave some cellular proteins [156,157,158,159,160,161,162,163] indicating that the HIV-1 enzyme is active in the host cell. Amongst these cellular targets, one finds proteins involved in translation such as eIF4GI, PABP, and eIF3d [157,158,163]. Carrasco et al have shown in lymphoid cell lines infected with the HIV-1 NL4.3 strain that both eIF4GI and PABP were cleaved and this effect could be prevented through the addition of the specific HIV-1 protease inhibitor Saquinavir [157,163]. Two reports from different labs have shown that eIF4GI was a substrate for the HIV-1 viral proteases in in vitro systems using recombinant proteins. Unlike picornaviral proteases, which bisects eIF4G I and II into two main fragments [164], the retroviral protease targets two sites on eIF4GI that are located on either sides of the central binding domain of eIF3 and eIF4A [163,165]. Moreover, this is highly specific to eIF4GI as the eIF4GII isoform is not processed by the viral protease [165] although its expression level during infection may be reduced [163]. As a consequence of this proteolytic destruction, cap-dependent translation is strongly affected by the rupture of the cap-eIF4E-eIF4G-PABP interaction. However, translation initiation mediated by IRESs derived from the EMCV and HCV genomes is sustained [157,163,165,166,167]. Interestingly, translation of a reporter gene driven by the 5′ UTR of the unspliced HIV-1 mRNA was repressed, suggesting that eIF4GI was required for Cap/DDX3-dependent translation [166]. This contrasts with translation of a reporter construct driven by the HIV-1 5′ UTR followed by the Gag coding region, which was not affected, and even stimulated, by the addition of the enzyme [163,167]. These data suggest that the Gag coding region plays a role in the translation of the HIV-1 mRNA and this will be developed later in this review. 

Another viral protein, Vpr, has also been involved in the control of HIV-1 translation. Vpr has pleiotropic functions during infection since it participates in reverse transcription and nuclear import, LTR transactivation, apoptosis, and cell cycle progression [168]. Since 1995, it has been known that the expression of Vpr could impose a cell cycle arrest in the G2 phase [169,170,171,172]. The mechanism by which Vpr can block the cell cycle is not fully understood but appears to be linked to the activation of the DNA damage response pathways and involves several interactions of Vpr with the ATR (ataxia-telangiectasia and Rad3-related) protein, the ubiquitine E3 ligase and the SLX4 complexes [173]. Above all, the G2/M phase arrest of the cell cycle was demonstrated to be associated with a strong decrease of cap-dependent translation [145,174,175,176] which results mostly from the phosphorylation of 4B-BP and the sequestration of eIF4E [174]. During this phase, translation mediated by IRESes was shown to be not affected as it does not require eIF4E to initiate [177,178,179]. Consistent with eIF4E sequestration, cellular mRNAs are less associated with polysomes in HIV-1-infected cells, which confirms the inhibition of cap-dependent translation initiation [144,145,180]. Indeed, expression of the Vpr protein alone is sufficient to promote the dephosphorylation of both 4E-BP and eIF4E [145]. Despite this repression, the translation level of the HIV-1 unspliced mRNA is not affected [145], indicating that alternative mechanisms of ribosome recruitment exists.

#### 5.2.2. The HIV-1 5′ UTR Promotes Internal Initiation

Viral IRESs are generally long and structured RNA regions that can interact with PIC components; this is generally evidenced by the use of a bicistronic vector in which the putative IRES sequence is placed in the intercistronic region. Brasey and colleagues have demonstrated in HeLa cells that the 5′ UTR of the unspliced HIV-1 genomic RNA exhibits IRES activity [181]. The minimal segment for function was mapped to a ≈225 nt long RNA region that overlaps the PBS, DIS, SD, and Psi RNA signals (Figure 3) [181]. The recruitment of the ribosome on this IRES was rather inefficient in the RRL unless cellular extracts were added [182] reflecting the need for additional cellular factors. Since the initial manuscript from Sonenberg and colleagues, many other reports have validated the IRES activity of the HIV-1 5′ UTR in HeLa cells [181,182,183,184,185,186], but also in T cells and monocytes [187,188,189]. Surprisingly, introduction of mutations and deletions within the IRES sequence failed to identify any individual RNA structures or motifs that would be critical for ribosomal recruitment [182,187,188] as it is usually the case for many other viral IRESs [37,190,191,192]. However, most data confirm that the PBS, SD, and DIS structures are required to sustain IRES activity [185,187,188,193]. Interestingly, an RNA sequence that lies immediately upstream to the PBS signal was identified as a negative regulator of internal ribosome entry and was called IRENE, short for IRES negative element [187]. Translation driven by the HIV-1 IRES is robust to the co-expression of picornaviral protease targeting eIF4G1 [181,194], indicating that eIF4E and the N-terminal domain of eIF4G are not required in a similar manner to many other viral IRESs. Finally, although internal initiation was essentially characterized by using laboratory-adapted viral strains (NL4.3, Lai, and Hxb2), recent studies have extended these results to sequences derived from natural clinical variants of HIV-1 [185,195]. 

Translation initiation can occur either at an AUG located immediately at the 3′ border of the IRES as in the case of EMCV, HCV, and CrPV IRESs, or several nucleotides downstream of the ribosome binding site as it is the case for the poliovirus IRES, and in this case, will require additional ribosomal scanning to reach the initiation site [31,196]. Thus, the question of ribosomal scanning after landing on the HIV-1 IRES was raised. By adding low concentrations of edeine, which interferes with scanning and recognition of the AUG codon, to an RRL supplemented with HeLa extracts, Carvajal et al. have shown an inhibition of translation driven by the PV IRES but not from the HCV IRES nor from the HIV1 5′ UTR [185], suggesting that initiation takes place with no scanning. In contrast with these data, Plank et al. have reported that HIV-1 IRES-mediated translation was sensitive to hippuristanol (a small molecule inhibitor of eIF4A) in Jurkat cells. Introduction of an additional AUG codon upstream of the natural initiation codon of Gag also inhibited translation at the authentic initiation codon, suggesting that scanning was required after the landing of the PIC at the 3′ border of the IRES [189]. Requirement for a scanning step was confirmed in vitro by the addition of a recombinant trans-dominant negative eIF4A^R362Q^ mutant [197] or hippuristanol to the RRL, which was sufficient to repress expression mediated by the HIV-1 5′ UTR [198].

#### 5.2.3. HIV-1 IRES *trans*-Acting Factors

The involvement of cellular proteins in HIV-1 translation was first evidenced by the fact that Hela cell extracts prepared from cells arrested in the G2/M phase were necessary to support initiation in vitro [181,182]. In addition, data obtained from structural studies using SHAPE showed a very different profile whether, or not, the probing was carried out in the presence, or absence, of Hela cell cytoplasmic extracts from cells synchronized in the G2/M phase. This showed a different pattern of RNA protection, notably on the TAR apical loop, the IRENE, and the DIS elements. Specific RNA protections were also shown inside the poly(A) the stem-loop, the PBS and the Psi domains with G2/M synchronized cell extracts [182]. These differences in the pattern of RNA protection suggests the binding of proteins that are specifically expressed during the G2/M phase of the cell cycle and which may control the activity of the HIV-1 IRES. A proteomic approach has also identified a set of 18 cellular proteins associated with the HIV-1 IRES [182], although their roles have not been further investigated. However, transcription and expression of hnRNP A1 was shown to be enhanced during HIV-1 infection and this was accompanied by a relocation of the protein from the nucleus to the cytoplasm when the Rev protein was expressed [183]. Silencing of hnRNP A1 reduced expression mediated by the 5′ UTR IRES in the context of bicistronic reporter genes and this effect could be restored via co-expression with a si-RNA resistant form of the protein [183]. As such, hnRNP A1 is the first ITAF described for the HIV-1 5′ UTR IRES, although its precise function remains to be determined. Two other additional proteins have been involved in the regulation of HIV-1 IRES activity: the first one is the Elav-like protein HuR, which inhibits IRES activity [184], and the other one is the S25 small ribosomal protein, which was shown to promote recruitment of the PIC on the IRES although no direct interaction between S25 and the viral RNA could be evidenced [185]. Interestingly, S25 was previously described for a similar role on the EMCV and HCV IRESs [199]. We have shown that DDX3 interacts with the TAR stem loop but also with regions of the 5′ UTR IRES [141], and this characteristic opens the possibility that DDX3 may participate in IRES-mediated initiation. Consistent with this hypothesis, this is not a surprise that the silencing and/or over-expression of DDX3 in 293T cells modulates the activity of the HIV-1 5′ UTR IRES in the context of a bicistronic vector [186]. Finally, the activity of the HIV-1 IRES is strongly stimulated in lymphocytes and dendritic cells compared to Hela [188], suggesting a cell type specificity for viral translation that could reflect the need for additional specific *trans*-acting factors that remain to be determined. 

#### 5.2.4. Role of the Viral Proteins in HIV-1 Translation Mediated by the 5′ UTR

Packaging of the genomic RNA is one of the last stages of viral infection and begins with the interaction of the neo-synthesized Gag polyprotein with the encapsidation signal (Psi) located in the 5′ UTR [200,201]. As Psi overlaps with the viral IRES, the nucleation of Gag interferes with incoming ribosomes that initiate translation, both from the 5′ cap and the IRES. Anderson et al. have conducted in vitro experiments in which translation mediated by the 5′ UTR was assayed in the presence of increasing amounts of recombinant Gag protein. They could show that Gag exerted a dual opposite effect on HIV-1 translation: it was stimulatory at low doses but inhibitory at high concentrations of the recombinant proteins and this was strictly dependent on the presence of the RNA binding domain of Gag [202]. This suggests that the binding of the Gag polyprotein to its cognate mRNA serves to regulate its own translation. 

The regulatory Rev protein has also been involved in HIV-1 translation. Indeed, it was shown that Rev can interact with the A loop of the DIS structure [150,151] and is able to modulate translation of mRNAs that harbor the HIV-1 5′ UTR in a dose-dependent manner [150,203]. Low quantities of Rev specifically stimulate the ribosome recruitment on the HIV-1 mRNA while high doses inhibit translation [150].

As previously mentioned, the HIV-1 protease inhibits cap-dependent but not IRES-mediated initiation driven by the EMCV IRES [163,166,167]. Surprisingly, the impact of the HIV-1 protease on its cognate IRES has never been characterized in the context of bicistronic mRNAs. Expression directed by the HIV-1 5′ UTR from a monocistronic reporter mRNA was affected by the cleavage of both eIF4G1 and PABP [166]. However, when the reporter construct was extended to include the Gag ORF downstream from the HIV-1 5′ UTR, then translation was resistant to expression of the viral enzyme pointing out to a regulatory role of the Gag coding region [163,167].

#### 5.2.5. The Role of the Gag Coding Region

Seminal studies have shown that the Gag coding region could exert a negative effect on the activity of the 5′ UTR IRES [181,188,204] and several explanations have been proposed. First, the region coding for the matrix within the Gag ORF contains instability elements named INS-1 that negatively impact on HIV-1 IRES-driven translation, both in in vitro and in ex vivo experiments [205]. Although the mechanism remains unclear, the overexpression of Rev or hnRNP A1 can alleviate this inhibition. Second, potential RNA–RNA interactions occurring between the Gag coding region and the 5′ UTR could change structural motifs that are essential for IRES activity. Indeed, the 5′ UTR of the unspliced mRNA can adopt two alternative structures: a long-distance interaction and a branched multiple-hairpin structure in which the AUG codon is differentially exposed. However, no major differences in translation initiation has been reported from these two structures [187,188,206], but these studies were performed with a few nucleotides downstream from the AUG codon and the impact of the entire matrix coding region has not really been analyzed at the level of the RNA structure.

Interestingly, the HIV-1 matrix coding region can promote internal initiation of the ribosomes, both in vivo and in vitro [204], and this mechanism was found to be conserved in the type 2 human and simian immunodeficiency virus relatives [207,208,209]. This was initially demonstrated in poliovirus-infected cells that expressed bicistronic constructs containing the Gag coding region in the intergenic spacer. As a result, Gag-p55 was expressed efficiently together with a shorter isoform of 40 kDa, which corresponds to an N-terminal truncated isoform of Gag produced from an internal initiation event. Since then, the expression of Gag-p40 and IRES activity have been largely studied in several in vitro and in cellulo translation assays [198,204,210,211,212], and was also found in sequences derived from HIV-1 infected patients [211]. Interestingly, the binding of a 40S ribosomal subunit can occur directly to the Gag coding region in the absence of any eIFs in filter binding assays [212]. By using SHAPE structural modeling coupled with functional translational and biochemical assays, two independent internal ribosomal binding sites have been described (Figure 3): the first site is immediately downstream from the authentic AUG initiation of Gag-p55 while the second is in the vicinity of the AUG-p40 codon [212]. Thus, the presence of site 1 is consistent with a backward shift of the PIC to the AUG-p55 codon to promote translation of Gag-p55 [198,210,212]. Consistent with IRES general features, translation initiation on the Gag IRES is robust to picornaviral proteases and cap analog treatments [198,213], but it requires the eIF4A helicase for AUG recognition [198]. 

#### 5.2.6. Adenosine Methylation Regulate Future of the Viral Transcripts 

HIV-1 infection is associated with an increase of adenosine methylation of both cellular and viral mRNAs [214,215,216] that drives opposite effects on viral expression and replication. Lichinchi et al. have characterized 14 distinct sites of methylation found in the 5′ UTR, the coding regions, and within the RRE element. The silencing of the two METLL3 and METLL4 writer proteins or the ALKBH5 eraser significantly decreases and increases HIV-1 replication, respectively, indicating that m^6^A modifications are needed. A focus on the impact of m^6^A residues located in the RRE showed that methylation controls the interaction between RRE and the Rev protein to enhance nuclear export of unspliced and partially spliced mRNAs. m^6^A residues were found in the sequence spanning nucleotides 170 to 420 (from the +1 of transcription of the LAI strain) that include the PBS, DIS, and Psi elements, and the beginning of the Gag coding region, but their functions have not yet been characterized [214]. Kennedy et al. have highlighted m^6^A editing only in the last 1.4kb of the unspliced mRNA. This difference with the work of Lichinchi and collaborators could be attributed to the use of different kinds of cells (MT4 T vs CEM-SS cells) or HIV-1 strains (LAI vs NL-4.3), but also from differences in experimental settings. These methylated adenosines interact with the reader proteins (YTHDF 1 to 3) in infected cells, and overexpression of the YTHDF proteins correlates with an enhancement of both viral transcription and viral proteins expression [215]. Tirumuru and collaborators have also characterized m^6^A residues in both UTRs, in the Rev reading frame and the Gag coding region, and all these are recognized by the YTHDF reader proteins [216]. In this study, they also show that Gag protein production was stimulated by the presence of m^6^A residues on the unspliced mRNA. Change in the expression of writers and/or eraser proteins modulates the viral mRNAs’ translation [216]. They also showed that the YTHDF proteins are implicated at different levels of the viral replication cycle, and they negatively inhibit the viral reverse transcription step and promote HIV-1 gRNA degradation [216,217]. Consistent with this last observation, the impact of the interaction between m^6^A residues and the YTHDF family also negatively regulates the replication, but not at the translational level, of other RNA viruses such as HCV and Zika virus (see these excellent revues that compare m^6^A modifications from several RNA viruses: References [218,219]).

The identification of methylated adenosines inside the HIV-1 unspliced mRNA has just emerged and these nucleotide modifications appear to differentially regulate expression of the unspliced mRNA, but also other steps of the viral replication. Changes in the expression pattern of the writer and eraser proteins have consequences on viral protein level and infectivity [215,216]. In contrast, reader proteins, which interact with the m^6^A residues, most probably address the viral RNA to either a translational or degradation pathway, but the mechanistic and determinants remain to be characterized. 

## 6. Translation of the HIV-1 spliced mRNAs

All viral mRNAs share a common region of 289nt from the +1 transcription to the major splice donor site, which includes TAR, poly(A), PBS, and DIS RNA elements. Thus, alternatively spliced transcripts differ by the RNA segment comprised between the SD and their cognate AUG codon; as a consequence, they exhibit 5′ UTRs with different lengths and structures [88,102,220,221], which certainly modulate translational efficiency. Surprisingly, a global systemic investigation of how protein synthesis is controlled in spliced viral mRNAs has never been undertaken. 

The knock-down of DDX3 expression in HIV1 (NL-4.3) infected Jurkat T cells’ reduced translation of Gag-p55, but also that of Nef and Vif proteins without affecting their cytoplasmic mRNA level [142]. Similarly, expression of the Tat and Rev mRNAs in cellular extracts depleted in DDX3 were found to be less associated with the polysomal fraction in comparison to non-depleted controls [222]. These two reports strongly suggest that DDX3 could be required for translation of spliced mRNAs, probably due to the presence of the TAR RNA element at their 5′ end. However, studies performed by Kieft et al. showed that the 5′ UTR derived from transcripts coding for Nef, Vif, Vpr, and Vpu are able to promote expression of the second cistron in a bicistronic RNA context in Jurkat T cells [188]. This is not surprising as all 5′ UTRs are composed of a common component (PBS-DIS-DS) that exhibits IRES activity. In that sense, the variable RNA segment that lies 3′ to the border of the IRES, and which is specific to each individual transcript, would only be a modulator in the process [189]. Consistent with these data, the 5′ UTR of the two major transcript isoforms coding for Tat can also mediate internal initiation in a bicistronic context [223]. Therefore, it appears that spliced transcripts exhibit common features with the unspliced genomic mRNA with the ability to use internal ribosome binding or ribosomal scanning from the 5′ end. 

The reading frame coding for the Env protein is the only ORF that is not preceded by a splice acceptor site. As a consequence, and due to differential alternative splicing, 16 natural bicistronic mRNAs are produced. All of them are composed of the Vpu ORF followed by Env but they differ by their 5′ UTR [1,220,224]. Translation initiation of the Vpu/Env bicistronic mRNA is mediated by a leaky scanning mechanism, which is controlled by the presence of the short ORF located at the initiation start codon of Vpu (AUGUAAUG, Vpu-AUG codon is underlined). The Vpu-AUG codon, which lies in an unfavorable context, is poorly recognized and most of the ribosomes scan down to the downstream Env-AUG codon [225,226]. A minority of the bicistronic mRNAs contains an additional translation start codon upstream of the Vpu ORF [1,220,221,224] that abrogates translation initiated at the Vpu-AUG codon, but not at the Env-AUG codon [224]. Curiously, mutations of this upstream AUG codons of Rev, alone or in combination with the mutation of the Vpu-AUG codon, do not impact the expression of the Env protein, which is not consistent with a leaky scanning mechanism [224]. Thus, this indicates that expression of the Env protein could also be driven by another alternative initiation mechanism that remains to be determined.

## 7. Concluding Remarks

In contrast to other viruses, such as HCV, EMCV, or poliovirus, for which translation relies exclusively on IRES-mediated initiation, the unspliced HIV-1 genomic RNA can use three different mechanisms of ribosomal recruitment:

(i) Similar to cellular mRNAs, ribosome attachment can occur at the cap structure and it requires different RNA helicases (RHA, DDX3, La, and TRBP) to promote unwinding of the 5′ end TAR structure. However, how these RNA helicases cooperate in the course of the viral cycle remains a question to be addressed. Similarly, how the cap structure of the viral transcripts is recognized and which cellular proteins intervene in the process is also a matter of debate. Indeed, during viral infection, translation of cellular mRNAs is reduced due to the sequestration of eIF4E by 4E-BP induced by Vpr, which renders eIF4E unavailable [145]. Thus, two large protein complexes composed of DDX3/eIF4G/eIF4A/eIF3 and CBP80/Rev/eIF4A can substitute for eIF4E to promote initiation at the cap structure. The cellular modifications triggered by infection certainly contribute to a subtle interplay between all possible mechanisms of translation initiation at the 5′ end of the mRNAs. Although a great amount of progress has been made towards understanding translational regulation of HIV-1, only a few studies addressed this question in the context of viral replication. This will certainly be one of the challenges for the next few years.

(ii) The PIC can also interact with the IRES in the 5′ UTR and the IRES in the matrix coding region. Here again, the cellular and viral proteins involved in these processes are not fully understood. Technically, it is very difficult to measure the ratio of ribosomes delivered to the cap structure versus ribosomes that are loaded onto the IRES. This may be possible in the future through the development of inhibitors specifically targeting only one of the two mechanisms (cap or IRES). Interestingly, as both DDX3 and Rev can bind to structures located in the IRES [141,147,150,151], it suggests that they could also be involved in both mechanisms. This opened question has to be resolved.

(iii) Finally, an additional level of complexity came with the discovery of the recent implication of methyladenosines that can also promote direct ribosomal entry. These modifications could also modify the RNA structure [227] and may impact the IRES activity. The link between methylation of the adenosine and translation driven by IRES remains to be showed, but these methylated adenosines on the viral RNA modulate both mRNA stability and mRNA translation efficiency, which may provide new avenues for novel potential antiviral targets.

In conclusion, translational control of HIV-1 is a very complex process with multiple strategies to utilize or detour the cellular machinery. Although a lot of work has already been done to decipher some of the molecular mechanisms, some issues remain to be addressed and future work should clarify how the virus juggles with all these mechanisms during the course of viral replication. 

## Figures and Tables

**Figure 1 ijms-20-00101-f001:**
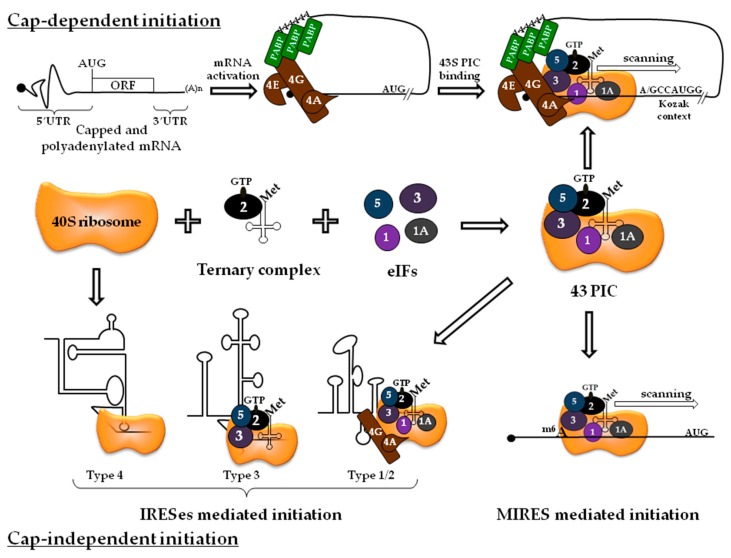
Model for cap-dependent and -independent ribosome recruitment on mRNAs. Interaction of the ternary complex to the 40S ribosome is enhanced in the presence of eIFs to form the 43S pre-initiation complex (PIC). 43S PIC can be recruited on activated mRNAs by the eIF4F complex composed of the eIF4E cap binding protein, the eIF4G scaffold protein, and the eIF4A helicase. 43S PIC bound to the mRNA scans the 5′ untranslated region (UTR) to initiate translation at the AUG initiation codon. This is the cap-dependent translation initiation mechanism. Cap-independent ribosome attachment is largely mediated by IRESes in which components of the 43S PIC can interact directly with some RNA structures. The requirement of eIFs to promote ribosome attachment depends of the type of the IRES: briefly, type 1 and 2 IRESes require all eIFs except eIF4E, type 3 IRESes do not need any of the eIF4F complex components, and type 4 IRESes do not rely on any eIFs nor Met-tRNAi. Finally, the 43S PIC can bind m^6^A located into the 5′ UTR.

**Figure 2 ijms-20-00101-f002:**
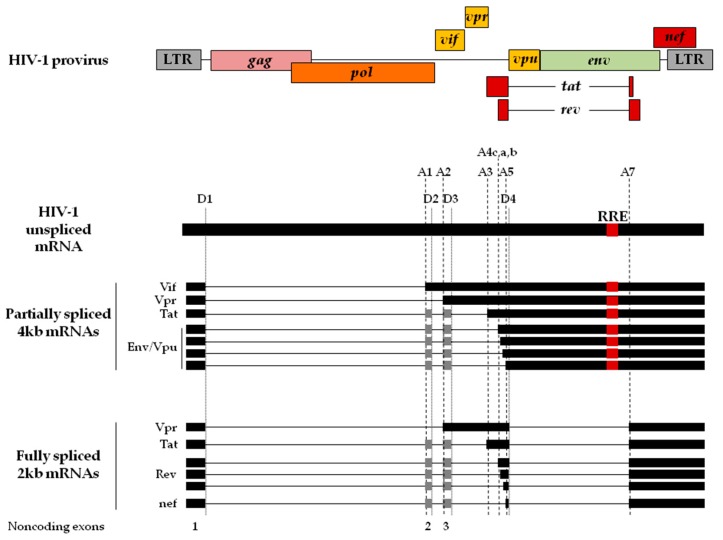
Schematic representation of the HIV-1 genome. The different colored boxes indicate the HIV-1 ORFs located between the two LTRs. Position of the conserved donor (D1 to D4) and acceptor (A1 to A7) sites are shown. The HIV-1 primary transcript serves as both mRNA that encodes for the Gag and Gag-Pol polyproteins and as genomic RNA. The red RRE box corresponds to the binding site of Rev, which is required for the nuclear export of partially spliced and unspliced RNA transcripts. The exons of the partially and fully spliced mRNAs are shown in black boxes. The first exon is a noncoding exon (exon 1) present in all HIV-1 transcripts. Either both or one of the small noncoding exons 2 and 3 (grey boxes) are included in a fraction of the corresponding mRNA species leading to the diversity of the HIV-1 mRNAs.

**Figure 3 ijms-20-00101-f003:**
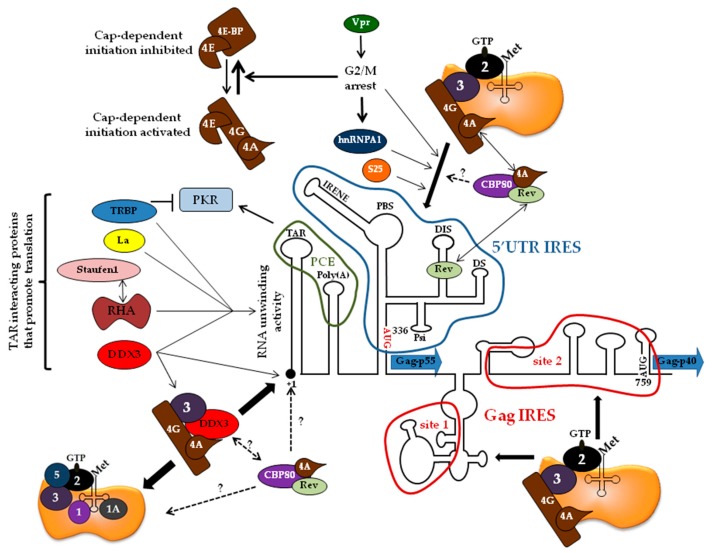
Translation initiation of the HIV-1 unspliced mRNA. Scheme of the RNA structures of the first ≈800 nt, which includes the 5′ UTR composed of the TAR, poly(A), PBS, DIS, SD, and Psi element required for viral replication, and the beginning of the Gag-coding region. The cap structure is at position +1 and the Gag-p55 and Gag-p40 initiation codons are located at positions 336 and 759, respectively. Cap-dependent translation is inhibited by the presence of the TAR element and needs the TAR interacting proteins (such as TRBP, La, Staufen1, RNA helicase A (RHA), and DEAD-box polypeptide 3 (DDX3)) to certainly promote RNA unwinding and ribosomal recruitment at the 5′ end of the mRNA. Two complexes composed of DDX3, eIF4G, eIF4A, and eIF3 on the one hand, and cap binding protein 80 (CBP80), Rev, and eIF4A on the other hand exist and could drive ribosome recruitment on the mRNA independently of eIF4E. The 5′ UTR IRES is depicted inside the blue line, and ribosome attachment can be promoted via hnRNPA1 and S25 proteins and is enhanced during cell cycle arrest in the G2 phase, which is induced by Vpr and induces the sequestration of eIF4E by 4E-BP. The two internal ribosome binding sites of the Gag-coding region are indicated inside the red lines and they drive expression of Gag p55 and Gag p40. The PCE and IRENE regions are also indicated.

**Table 1 ijms-20-00101-t001:** The four main classes of viral IRESs according to their requirement for eIFs (non-exhaustive list).

IRES Type	eIFs Required	Family	Genus	Species
I	eIF4G, eIF4A, eIF2, eIF3, eIF1, eIF1A, eIF5	Picornaviridae	Enterovirus	Poliovirus [25]
				Rhinovirus A [34]
				Human coxsakievirus B3 [35]
II	eIF4G, eIF4A, eIF2, eIF3, eIF1, eIF1A, eIF5	Picornaviridae	Cardiovirus	Encephalomyocarditis virus [26]
			Aphtovirus	Foot-and-mouth disease virus [36]
III	eIF2, eIF3, eIF5	Flaviridae	Hepacivirus	Hepatitis C virus (HCV) [37]
			Pestivirus	Classical swine fever virus [37]
		Picornaviridae	Senecavirus	Seneca Valley virus [38]
			Teschovirus	Porcine teschovirus 1 [39]
			Enterovirus	Porcine enterovirus 8 [40]
			Sapelovirus	Simian picornavirus 9 [41]
		Dicistroviridae	Cripavirus	Drosophila C virus [42]
IV	none	Dicistroviridae	Cripavirus	Cricket paralysis virus [43]
			Apavirus	Taura syndrome virus [44]

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
