# Peer review of "Focus on Translation Initiation of the HIV-1 mRNAs"

_ijms, 2018, doi:10.3390/ijms20010101_

Reviewer 1 Report

In this review, the authors, have first described  the general mechanism of translation initiation (cap-dependent or cap-independent) and then, they focused on mechanisms of HIV-1 mRNA translation initiation. In my opinion, this review is interesting, well structured and balanced. However, I suggest the authors to extend this review with a new chapter dealing mechanisms of HIV-1 mRNA protein synthesis involved in HIV-1 activation/latency/re-activation process.

Author Response

Comment: In this review, the authors have first described the general mechanism of translation initiation (cap-dependent or cap-independent) and then, they focused on mechanisms of HIV-1 mRNA translation initiation. In my opinion, this review is interesting, well structured and balanced. However, I suggest the authors to extend this review with a new chapter dealing mechanisms of HIV-1 mRNA protein synthesis involved in HIV-1 activation/latency/re-activation process. 

Response: We thank the reviewer for its positive comment. Although this is a great suggestion, we prefer to remain focused outside the 'activation/latency/re-activation' process of HIV-1 as it covers a lot of work published and would almost merit a review on its own....

Reviewer 2 Report

It is an interesting and timely review, comprehensibly discuss the host as well as viral mRNA regulation and protein synthesis, with a particular focus on HIV-1. This article may be a fit in the Viruses journal, if considered. Following helpful comments are mean to improve the paper. Figure quality and text can be improved.

1. Abstract: line 15: “some viral proteins also regulate..PKR” what are those viruses or viral proteins? Please indicate a group of or a few viruses because the title is focusing HIV-1.

2. The first citation appears in page 2! I would recommend citing relevant articles for referring facts, for example: “HIV-1 produces 50 RNAs and 15 proteins”

3. English check, especially punctuations.

I am not sure about the missing commas (“,”) before “which” in many sentences, a few examples:

Page 1 lines 15-16, 41; page 2 line 59... and more instances

And in series, for example Vpu, Vpr ans Vif)

Grammar-proofreading from a native-speaker will certainly help! Some examples.

Abstract, line 12: ...adenosine methylation used by ribosomes to translate....?

Page 3 line 96 “decrease of the recognition efficiency of the start codon”

Page 2 line 99-100: and allows..

Page 7 line 249: thorough fully

4. Enhance the figure quality. An advice for writing on cartoons: In Fig 1 (and Fig 2, CBP80), eIF3 is (represented as small purple circles) rightly marked in white on purple background. But for other factors PABP (tiny and not clear), eIF2, 4E, 4G, 4A are given in black on the red or green background, which is hardly evident.  Please make it easily readable, perhaps write out of cartoons, if required. Also indicate 5’ and 3’ UTR in the RNA stretch. Increase font size: m6A modification and Kozak sequence marked in the RNA are obscure. I can’t find MIRES in here (as referred in page 5, line 150)

It is m6A, and not m6A! Also N7, please correct throughout the text and figure.

5. Introduction can be written in a way to invite readers and give a clue about what are the concepts about to follow.

6. IRES part of text and table: Authors need to think about relating this with HIV-1 scenario or think about giving appropriate article title. Or please inform the readers that HIV-1 IRES part follows .

7. Be more specific and add more detail on m6A/viral epitranscriptomics: what are the key methyl transferease involved in the catalysis of methylation? Name some demethylase and discuss how they cooperate (in viral context). What are the consequences of m6A in different viruses (please look for HIV, IAV, HCV and other flaviviruses), are they always promoting translation? What is the sequence specificity for m6A methylation to happen (or is it random? there must be a selectivity of adenines to get methylated). There are some recent, great reviews out there (Kennedy et al 2017, Horn and Sarnow 2017), please refer and discuss them. I would also be cautious to state that if m6A is mediating translation or it is keeping RNAs from decay (enhancing stability, thus having more RNAs destined to produce more proteins), considering the mechanistic aspects are not fully understood and the field is at its infancy.

8. Section 2.3 in page 5 contains only two citations. Are they sufficient to vouch the content?

9. Page 5 line 196 and elsewhere: 9700pb, referring 9700bp?

10. For the sake of readers’ best understanding, it is important to sketch an additional figure depicting the genome organisation of HIV-1 and as a sub-figure indicate different splicing patterns (sites) of transcripts that produce various HIV-1 proteins.

11. Please add a specific citation in page 6, line 230, together with/insead of reference 73

12. Page 9: Does Staufen 1 known to interact with HIV-1 TAR for translation? Please include and discuss this. Staufen 2 was recently reported to have a role in RNA export, see Banerjee et al 2014 10.1186/1742-4690-11-18

13. Page 9 line 355-59: here authors raise an argument or rationalizing what could happen. It is better to write the possibilities as “several mechanisms can take place”. 

Author Response

Point 1: Abstract: line 15: “some viral proteins also regulate. PKR” what are those viruses or viral proteins? Please indicate a group of or a few viruses because the title is focusing HIV-1.

 Response 1: The sentence in the abstract has been modified to lead the reader only on HIV-1 and avoid misunderstanding

Point 2: The first citation appears in page 2! I would recommend citing relevant articles for referring facts, for example: “HIV-1 produces 50 RNAs and 15 proteins”

 Response 2: Citations have been added on the first two pages at lines 38 and 63.

Point 3: English check, especially punctuations.

 Response 3: We carefully read the manuscript and did the suggested corrections and others.

Point 4: Enhance the figure quality. An advice for writing on cartoons: In Fig 1 (and Fig 2, CBP80), eIF3 is (represented as small purple circles) rightly marked in white on purple background. But for other factors PABP (tiny and not clear), eIF2, 4E, 4G, 4A are given in black on the red or green background, which is hardly evident.  Please make it easily readable, perhaps write out of cartoons, if required. Also indicate 5’ and 3’ UTR in the RNA stretch. Increase font size: m6A modification and Kozak sequence marked in the RNA are obscure. I can’t find MIRES in here (as referred in page 5, line 150)

 It is m6A, and not m6A! Also N7, please correct throughout the text and figure.

 Response 4: The quality of all the figures (colours, texts and cartoons size) has been improved to be of better quality for the reader. We can provide to the journal the initial PowerPoint document if it can help for the image resolution.

All m6A have been corrected in the manuscript and in the figures.

Point 5 and 6:

Introduction can be written in a way to invite readers and give a clue about what are the concepts about to follow.

IRES part of text and table: Authors need to think about relating this with HIV-1 scenario or think about giving appropriate article title. Or please inform the readers that HIV-1 IRES part follows .

 Response 5 and 6: The end of the introduction has been rewritten to indicate to the reader the guiding principle of the review with an emphasizing on the IRES part.

Point 7: Be more specific and add more detail on m6A/viral epitranscriptomics: what are the key methyl transferease involved in the catalysis of methylation? Name some demethylase and discuss how they cooperate (in viral context). What are the consequences of m6A in different viruses (please look for HIV, IAV, HCV and other flaviviruses), are they always promoting translation? What is the sequence specificity for m6A methylation to happen (or is it random? there must be a selectivity of adenines to get methylated). There are some recent, great reviews out there (Kennedy et al 2017, Horn and Sarnow 2017), please refer and discuss them. I would also be cautious to state that if m6A is mediating translation or it is keeping RNAs from decay (enhancing stability, thus having more RNAs destined to produce more proteins), considering the mechanistic aspects are not fully understood and the field is at its infancy.

 Response 7: The reviewer is right; this field of m6A/viral epitranscriptomics is at its infancy and remains not clear for the community with opposite effects on HIV-1. For this reason, we did not want to go too deep into the effect of m6A residues on HIV-1 translation and replication. We wanted to inform the reader of this incoming new field in order to guide them to adequate, complete and recent revues; this is now done by citing the works of Kennedy and Sarnow. Nevertheless, we have also included new elements into the part 2.2.2 in which we describe in more detail the writer and eraser family but also the consensus sequence and the modification rate. In addition, we now mentioned that, in contrast to HIV-1, m6A modifications can negatively affect the replication of others viruses like HCV and Zika virus.

Point 8: Section 2.3 in page 5 contains only two citations. Are they sufficient to vouch the content?

 Response 8: The reviewer is right and several citations have been added.

Point 9: Page 5 line 196 and elsewhere: 9700pb, referring 9700bp?

 Response 9: This has been corrected

Point 10: For the sake of readers’ best understanding, it is important to sketch an additional figure depicting the genome organisation of HIV-1 and as a sub-figure indicate different splicing patterns (sites) of transcripts that produce various HIV-1 proteins.

 Response 10: A figure describing the genome organisation and the several transcripts produced by alternative splicing has been added to the manuscript.

Point 11: Please add a specific citation in page 6, line 230, together with/insead of reference 73

 Response 11: specific citations have been added.

Point 12: Page 9: Does Staufen 1 known to interact with HIV-1 TAR for translation? Please include and discuss this. Staufen 2 was recently reported to have a role in RNA export, see Banerjee et al 2014 10.1186/1742-4690-11-18

 Response 12: We thank the reviewer for this good remark. The involvement of Staufen2 is now mentioned in the parts 4 and 5.1.

Point 13: Page 9 line 355-59: here authors raise an argument or rationalizing what could happen. It is better to write the possibilities as “several mechanisms can take place”. 

 Response 13: The sentence has been modified indicating that several mechanisms can take place.

Reviewer 3 Report

This manuscript comprehensively reviewed the processes involved in HIV translation initiation and viral and host factors that are involved in it. The content is very complete and and the coverage is extensive. Overall the organization of the review is also set in a way that would help improve the understanding of the paper. 

One suggestion is to improve the quality of the figures in the manuscript. While the quality of the text of the review is good, that of the figures can be improved. For examples the colored circles should be clearly labeled by their individual protein names instead of labeling a group of them as "eIFs". Also in figure 1 the ternary complex of translation initiation should indicate where the Met is on the tRNA.

Small errors:

Line 173, need a reference for the statement. 

Line 221, 9700pb should be 9700bp

Line 251, alters should be alter

Line 282, this is occurs should be this occurs

Author Response

Point1: This manuscript comprehensively reviewed the processes involved in HIV translation initiation and viral and host factors that are involved in it. The content is very complete and and the coverage is extensive. Overall the organization of the review is also set in a way that would help improve the understanding of the paper.

 One suggestion is to improve the quality of the figures in the manuscript. While the quality of the text of the review is good, that of the figures can be improved. For examples the colored circles should be clearly labeled by their individual protein names instead of labeling a group of them as "eIFs". Also in figure 1 the ternary complex of translation initiation should indicate where the Met is on the tRNA.

 Response 1: We thank the reviewer for its positive comment and he is right concerning the quality of the figures. They were modified to be more readable and understandable.

Point 2: Small errors:

Line 173, need a reference for the statement.

Line 221, 9700pb should be 9700bp

Line 251, alters should be alter

Line 282, this is occurs should be this occurs

 Response 2: all small errors have been corrected.